# The Synchronized Progression from Mitosis to Meiosis in Female Primordial Germ Cells between Layers and Broilers

**DOI:** 10.3390/genes14040781

**Published:** 2023-03-23

**Authors:** Yuxiao Ma, Wenhui Wu, Yun Zhang, Xuzhao Wang, Jiahui Wei, Xiaotong Guo, Man Xue, Guiyu Zhu

**Affiliations:** Shandong Provincial Key Laboratory of Animal Biotechnology and Disease Control and Prevention, College of Animal Science and Technology, Shandong Agricultural University, Tai’an 271000, China; 2020110460@sdau.edu.cn (Y.M.);

**Keywords:** chicken, primordial germ cells, proliferation, mitosis, meiosis

## Abstract

Layer and broiler hens show a dramatic difference in the volume and frequency of egg production. However, it is unclear whether the intrinsic competency of oocyte generation is also different between the two types of chicken. All oocytes were derived from the primordial germ cells (PGC) in the developing embryo, and female PGC proliferation (mitosis) and the subsequent differentiation (meiosis) determine the ultimate ovarian pool of germ cells available for future ovulation. In this study, we systematically compared the cellular phenotype and gene expression patterns during PGC mitosis (embryonic day 10, E10) and meiosis (E14) between female layers and broilers to determine whether the early germ cell development is also subjected to the selective breeding of egg production traits. We found that PGCs from E10 showed much higher activity in cell propagation and were enriched in cell proliferation signaling pathways than PGCs from E14 in both types of chicken. A common set of genes, namely insulin-like growth factor 2 (*IGF2*) and E2F transcription factor 4 (*E2F4*), were identified as the major regulators of cell proliferation in E10 PGCs of both strains. In addition, we found that E14 PGCs from both strains showed an equal ability to initiate meiosis, which was associated with the upregulation of key genes for meiotic initiation. The intrinsic cellular dynamics during the transition from proliferation to differentiation of female germ cells were conserved between layers and broilers. Hence, we surmise that other non-cell autonomous mechanisms involved in germ-somatic cell interactions would contribute to the divergence of egg production performance between layers and broilers.

## 1. Introduction

Broilers and layers are commercial chickens that provide meat and eggs to the food industry. These two populations of chickens exhibit dramatic differences in organ development and function after multiple generations of intensive selection for rapid growth or egg production, respectively. As to the reproductive physiology, the typical layer can produce almost 300 eggs a year, whereas an average broiler breeder only lays half of them. Increasing the egg production capacity in broilers increases offspring numbers, thereby maximizing meat production. However, whether the intrinsic reproductive capacity of the broilers compared to the layers limits the egg production was not determined. The production of a chicken egg is a complicated, multistep process that involves coordinated interactions between the ovary, follicle, and oviduct. Nevertheless, the absolute number of available female germ cells may be one of the inherent factors that could contribute to the ovulation differences observed between broilers and layers. The population size of germ cells within a female ovary was precisely controlled by cell replication (mitosis) and cell differentiation (meiosis). In other words, once the germ cells stop mitosis and enter meiosis, the pool of female germ cells is fixed and cannot be increased anymore. Therefore, in this study, we analyzed the cellular dynamics of meiosis and mitosis of female germ cells during embryonic development, which would ultimately determine the reproductive potential in broilers and layers.

Primordial germ cells (PGC) are the primary, undifferentiated stem cell type that will differentiate towards both oocyte and sperm fates. The cell lineages can pass on genetic and epigenetic information from one generation to the next [1]. In most animals, PGCs originate in the peripheral region outside the embryo and migrate across the embryo to the gonads [2]. Avian PGCs first appear in the dorsal epiblast and, during gastrulation, move towards the anterior extra-embryonic and the germinal crescent [3]. At around HH10 [4], the PGCs enter the embryonic blood vascular system along with the embryonic regions [5]. In addition, around HH15, PGCs start leaving the blood vessels and colonizing in the developing gonadal anlage [6,7]. Gonadal sex differentiation commences at E6–6.5 (HH stages 29–30), indicating the differentiation of the supporting somatic cells into Sertoli cells in ZZ males and pre-granulosa cells in ZW females [8]. By E9.5, female and male gonads are morphologically and histologically disparate. In females, ovaries are known to develop asymmetrically, and only the left ovary develops a thick cortex where the germ cells are located, while the right one regresses [9,10].

In female chickens, upon colonizing the developing gonadal ridge at E9, the PGCs undergo rapid proliferation via mitosis to establish a germ cell pool [11]. A few days later, all germ cells in the left ovaries enter meiosis at around E15.5 and are arrested in meiotic prophase I as primary oocytes [12]. There are approximately 18,000 PGCs in the ovary by E14.5, and in contrast, a mouse has around 3000 dividing cells at E12.5 [13]. Previous research shows that because of this rapid proliferation, mouse PGCs increased their numbers by 50-fold from E9.5 to E12.5 [14]. However, the proliferation dynamics of PGCs in chicken gonads start from the sex differentiation toward the first meiotic arrest occurring in the female, which was not clearly demonstrated. The mitotic cycle of PGCs in mammals is driven by the sequential activation of different types of cyclin-dependent kinases or modulating transcription factors. Numerous studies suggest that PGCs have an increased requirement for DNA repair ability compared to their somatic cell counterparts [15,16,17].

However, little is known about the cellular dynamics and the underlying molecular mechanisms that govern the mitosis to meiosis transition in chicken PGCs. Two distinct chicken breeds were used in this study. One of them is Beijing You chicken, a yellow-feathered Chinese local chicken that can be used for meat and eggs [18,19]. These chickens have a unique appearance, excellent meat quality, and an annual egg production of around 110 to 130. The other is the Hy-Line chicken, which is an excellent layer breed with high egg production and an annual egg production of around 310–330. In the present study, we described the conservative mechanism of actively proliferating in broiler (Beijing You) and layer (Hy-Line) chicken PGCs. We found that transcription factors such as insulin-like growth factor 2 (*IGF2*), E2F transcription factor 4 (*E2F4*), and E2F transcription factor 6 (*E2F6*) may be the potential signals leading to the rapid proliferation of two different types of PGCs. We also revealed the enriched biological processes and signaling pathways of significant DEGs identified commonly between the layer and broiler PGCs isolated at E10 and E14. Therefore, this study may shed light on the mystery of PGCs characteristics from a new perspective in birds.

## 2. Materials and Methods

### 2.1. PGCs: Isolation and Culture

Fertilized eggs from Hy-Line chicken and Beijing You chicken (You chicken) were incubated in an incubator at 37.8 °C and 60% humidity. PGCs were isolated from the 10-day (E10) and 14-day (E14) female embryos of Hy-Line chicken and You chicken and cultured according to previously reported [20,21]. Briefly, the isolated PGCs were cultured and amplified in the medium of knockout Dulbecco’s modified Eagle’s medium (KO-DMEM, Gibco, New York, NY, USA, 10829018) using chicken embryonic fibroblast cells as feeder cells at 39 °C in 5% CO_2_ with saturated humidity. The KO-DMEM contained 10 ng/mL bFGF (Novoprotein, Suzhou, China, C044), 7.5% defined FBS (Solarbio, Beijing, China, P00032), 2.5% chicken serum (Solarbio, Beijing, China, S9080), 25 ng/mL human Activin-A (Novoprotein, Suzhou, China, C687), 1 × NEAA (Gibco, New York, NY, USA, 11140050), 1 × B-27 supplement (Gibco, New York, NY, USA, 17504044), 1 × GS nucleoside supplement (Millipore, Burlington, MA, USA, ES-008-D), 1 × antibiotic–antimycotic (Gibco, CA, USA, 15240062), 1 × Glu-taMAX (Gibco, New York, NY, USA, 35050061), 1.2 mM sodium pyruvate (Gibco, New York, NY, USA, 11360070), and 0.1 mM β-mercaptoethanol (Solarbio, Beijing, China, M8210).

### 2.2. Alkaline Phosphatase Staining

The cultured PGCs were washed twice with PBS to remove the culture medium. Alkaline phosphatase (AKP) staining working solution (TransDetect, Beijing, China, MA101) was added to the cells and incubated for 30 min in a 39 °C incubator. Then, the AKP staining working solution was discarded, and the cells were washed twice with PBS and observed under a fluorescence microscope.

### 2.3. Immunofluorescence Staining

The PGCs were first fixed in 4% PFA for 10 min, infiltrated with 0.1% Triton X-100 for 10 min, and then blocked with 10% goat serum for 60 min. After that, cells were incubated with primary antibodies in the blocking solution at 4 °C overnight and then with secondary antibodies at room temperature for 60 min. The primary antibodies used were anti-SSEA1 (DSHB, IA, USA, MC-480) and anti-DDX4 (Abcam, Cambridge, UK, ab13840). Secondary antibodies were Alexa Fluor 555-conjugated goat anti-mouse IgM (Solarbio, Beijing, China, K0055G) and Alexa Fluor 488-conjugated donkey anti-rabbit IgG (Invitrogen, Waltham, MA, USA, A-21206). Counterstaining of the nucleus was performed using DAPI (Solarbio, Beijing, China, C0060).

### 2.4. Cell Counting Kit-8 Assay

The PGCs were inoculated in a 96-well plate for 0, 24, 48, and 72 h, respectively, and then 10 μL of Cell Counting Kit (CCK-8) reagent (Solarbio, Beijing, China, CA1210) was added into each well and incubated for 2 h. The absorbance at 450 nm was detected by an automatic enzyme-linked immunosorbent assay system.

### 2.5. EdU Assay

5-ethynyl-2’-deoxyuridine (EdU, Beyotime, Shanghai, China, ST067) was dissolved in sterilized PBS at 2.5 mg/mL and stored at −20 °C. Before staining, the PGCs were incubated with 10 μM EdU dissolved in culture medium for 12 h. Cells were fixed in 4% PFA for 10 min, permeabilized with 0.1% Triton X-100 for 10 min, and then blocked in 10% goat serum for 60 min. After that, cells were incubated with anti-DDX4 (Abcam, Cambridge, UK, ab13840) in the blocking solution at 4 °C overnight and then with Alexa Fluor 488-conjugated donkey anti-rabbit IgG (Invitrogen, Waltham, MA, USA, A-21206) at room temperature for 60 min. After washing with PBS, cells were incubated with EdU staining buffer (1 mM CuSO4, 100 mM Tris, 100 mM ascorbic acid, and 10 mM fluorescent azide) for 30 min according to previous protocols [22]. Counterstaining of the nucleus was performed using DAPI (Solarbio, Beijing, China, C0060).

### 2.6. RNA Extraction

Total RNA was extracted from PGCs of Hy-Line chicken and You chicken at E10 and E14 using the picopure RNA Isolation Kit (Thermo Fisher Scientific, Waltham, MA, USA). The quality and quantity of the RNA were determined using NanoDrop 2000 (Thermo, Waltham, MA, USA). The RNA integrity number was assessed during the analysis.

### 2.7. RNA-Sequencing

High-throughput RNA sequencing (RNA-seq) was performed using illumina novaseq6000. RNA-seq reads were cleaned and aligned to the reference chicken genome (galGal7) release using hisat2 (version 2.1.0). The gene read counts were calculated by featureCounts30 (version 2.1.0). The FPKM value was used to measure the abundance of each transcript. The R programming language (version 4.1.2) was used to evaluate the expression mode of differentially expressed genes (DEGs). Genes with |Fold Change| ≥ 1.2 and *p* < 0.05 were identified as DEGs. The GO functional enrichment analysis was conducted by bioinformatics (http://www.bioinformatics.com.cn/ accessed on 1 July 2022). Genes with FPKM ≥ 1 at least in one sample were defined as detected genes.

### 2.8. Quantitative Real-Time PCR

Total RNA isolated from PGCs of Hy-Line chicken and You chicken at E10 and E14 was used for RT-qPCR. 500–800 ng of the total RNA was reverse transcribed into single-stranded cDNA using an RT kit (Yugong Biolabs, Jiangsu, China, EG15133S). Differential expression genes were selected, and the RT-qPCR was performed using reaction systems (Servicebio, Wuhan, China, G3321-05) on a LightCycler 96. The 2^−ΔΔCT^ algorithm was employed to estimate the relative expression level of each gene. Primers were designed using Primer 5.0, as shown in Appendix A.

### 2.9. Statistical Analyses

The data obtained from all tests were expressed as mean ± standard error. One-way ANOVA and post hoc Duncan’s multiple range tests were used to determine the differences between groups using SPSS 22.0 (SPSS, Inc., Chicago, IL, USA). Results were considered significant at the level of *p* < 0.05 and extremely significant at *p* < 0.01.

## 3. Results

### 3.1. Cultivation and Characterization of PGCs

PGCs were isolated from the gonads of 10-day female embryos (E10) and 14-day female embryos (E14) of the You and Hy-Line chicken breeds and cultured in vitro. To verify the germ cell identity of the cultured cells, we first probed the cells for the presence of alkaline phosphatase (AKP). We observed a strong signal of AKP staining in the cultured PGCs (Appendix A). Furthermore, immunofluorescence staining of two germ cell markers, SSEA-1 (stage-specific embryonic antigen 1) and DDX4 (dead box polypeptide 4), that have been widely used as germline stem cell markers to identify a germline, also confirmed the germ cell identity. In contrast, they were negative for chicken embryonic fibroblasts (Appendix A). These results demonstrated that we were able to isolate and culture PGCs while maintaining their germ-cell properties.

### 3.2. Comparison of the Proliferation Ability of PGCs from the Two Developmental Stages

We first compared the differences in proliferation between E10 and E14 PGCs in both chicken breeds. To quantify proliferation, we performed a CCK8 assay and found that E10 PGCs exhibited a higher proliferation rate compared to E14 PGCs (Figure 1a). We then performed an EdU assay and found that PGCs from the E10 stage had more EdU-positive cells and EdU-DDX4 double-positive cells than the E14 PGCs (Figure 1b,c). In addition, we also detected the expression of proliferation-related genes through RT-qPCR, and the results showed that in both breeds of chickens, the expression of the proliferation-promoting gene minichromosome maintenance protein 2 (*MCM2*) [23] was significantly lower at E14 than E10, while the expression of the proliferation-inhibiting gene Cyclin dependent kinase inhibitor 2A (*CDKN2A*) [24] was significantly higher at E14 (Figure 1d). These results show that PGCs from E10 undergo rapid proliferation and maintain strong mitotic competence through the cell cycle compared to the E14 PGCs in both Hy-Line and You chicken breeds.

### 3.3. RNA Sequencing of E10 and E14 PGCs

We found that the proliferation rate of PGCs gradually slowed down from E10 to E14, suggesting that PGCs are initiating meiosis from mitosis. To understand the molecular mechanisms of this process, we examined the global gene expression changes of PGCs from E10 to E14 in You chicken and Hy-Line chicken by RNA sequencing (RNA-seq). We performed RNA-seq using the Illumina novaseq6000 sequencing platform, with two biological replicates for each development stage. The CG content and average base quality met the basic requirements. The percentage of low-quality reads was less than 1%, and the percentage of Q30 bases was more than 90%. Detailed data are shown in Appendix A.

Pearson’s correlation coefficient (R) analysis between the replicates indicated that the R values within each group were 0.99, 0.97, 0.96, and 0.98, respectively, indicating high reproducibility between replicate samples (Appendix A). Principal component analysis (PCA) was used to explore the global changes in gene expression of PGCs at different developmental stages in You chicken and Hy-Line chicken (Appendix A). PCA analysis showed that samples at different stages were relatively well separated, while the replicates were closely clustered, indicating that the gene expression patterns of PGCs at different developmental stages were quite different.

### 3.4. Discovering Differential Co-Expression Genes Involved in Mitosis between You Chicken and Hy-Line Chicken

We identified a total of 936 significantly down-regulated genes and 1459 genes that were significantly up-regulated during the transition from E10 to E14 PGCs of Hy-Line chicken. Similarly, 1295 genes were significantly down-regulated and 1544 genes were significantly up-regulated in PGCs of You chicken during the transition from E10 to E14 (Figure 2a,b). It was found that in the two comparison groups, there were 1044 common differentially expressed genes (DEGs), including 425 co-upregulated genes and 209 co-downregulated genes in both strains of chicken (Figure 2c). The common DEGs were imported into the Gene Ontology (GO) databases for functional analysis (Figure 2d). The top biological processes enriched in these DEGs were the regulation of cell population proliferation, regulation of myeloid cell differentiation, immune response, and cell population proliferation. Further, GO analysis revealed that two biological processes“regulation of cell population proliferation” and “cell population proliferation” related to cell proliferationwere enriched. The genes corresponding to these cellular processes included activin A receptor type 2A (*ACVR2A*), colony stimulating factor 3 (*CSF3*), *IGF2*, and so on (Table 1). With the development of the ovary, *ACVR2A* and recombinant integrin α 1 (*ITGA1*) were up-regulated, and *CSF3*, *IGF2*, and ankyrin repeat domain 1 (*ANKRD1*) were down-regulated. These results indicate that *CSF3*, *IGF2*, and *ANKRD1* may be involved in the inhibition of PGC proliferation in both breeds.

Then, we selected eight representative DEGs that were involved in mitosis and cell proliferation and validated them by RT-qPCR (Figure 2e). First, high expression of interleukin-4 induced 1 (*IL4I1*), *CSF3*, prokineticin 2 (*PROK2*), *IGF2*, *E2F4*, and *E2F6* in E10 PGCs was verified. The results of fluorescence quantification validation showed good agreement with the RNA-seq results. In addition, cyclin-dependent kinase inhibitor 1B (*CDKN1B*) and adenovirus E1B 19 kDa protein-interacting protein 3 (*BNIP3*) genes associated with inducing apoptosis showed up-regulation in E14 PGCs in You chicken and Hy-Line chicken, consistent with the results of RNA-seq. Therefore, our RNA-Seq data was reliable and accurate. Meanwhile, these data demonstrated that these genes may play crucial roles during mitosis in coordinating the cell division cycle and ensuring the cell growth demands of any chicken breed.

### 3.5. Discovering Differential Co-Expression Genes Involved in Meiosis between You Chicken and Hy-Line Chicken

In chickens, all female germ cells in the left ovary enter meiosis at E15.5. To further define whether the genes expressed by PGCs at E14 have transitioned to the meiotic stage, we performed a Kyoto Encyclopedia of Genes and Genomes (KEGG) enrichment analysis of the 425 co-upregulated genes in You chicken and Hy-Line chicken (Figure 3a). Our analysis revealed that the up-regulated genes were associated with fatty acid processes and cell meiosis-related signaling pathways, such as “fatty acid metabolism”, “biosynthesis of unsaturated fatty acids” and “oocyte meiosis”. Genes corresponding to meiosis signaling pathways included structural maintenance of chromosome 3 (*SMC3*), cyclin-dependent kinase 2 (*CDK2*), and cell division cycle 20 homolog (*CDC20*) (Table 2). In addition to the above-mentioned genes, we focused on other genes such as those stimulated by retinoic acid 8 (*STRA8*) [25], retinaldehyde dehydrogenase 2 (*RALDH2*) [26], and RAD54-like (*RAD54L*) [27], which played important roles in meiosis initiation and were upregulated in E14 PGCs in both breeds.

Apart from this, several genes related to methylation modification were identified [28,29], among which Tet methylcytosine dioxygenase 2 (*TET2*) was significantly up-regulated in E14 PGCs of You chicken and Hy-Line chicken. Several studies have shown that *TET2* controls meiosis by regulating meiotic gene expression and plays multiple roles during meiosis and oocyte development [30]. Next, we validated the expression of *STRA8*, *RALDH2*, *RALD54L*, and *TET2* by RT-qPCR, and the result was consistent with the results of RNA-seq (Figure 3b). Therefore, the expression of meiosis-initiating genes and methylation-modifying genes together reveal that at E14, the PGCs of both You chicken and Hy-Line chicken are ready for meiosis initiation.

## 4. Discussion

PGCs of female chicken embryos colonize the gonads by day 6, after which they undergo rapid proliferation through mitosis and enter meiosis by day 15.5 [25]. The transition from mitosis to meiosis is important since the ultimate oocyte number is determined at this timepoint. This study reveals a synchronous progression from mitosis to meiosis in PGCs between broilers and layers by comparing proliferation rates and transcriptome analysis of PGCs from two different breeds of chicken.

We isolated and cultured PGCs in vitro and found that the proliferation rate of PGCs gradually declined from E10 to E14 in both You chicken and Hy-Line chicken. Using transcriptome analysis, we found the key genes related to proliferation that were co-expressed in the PGCs of both chicken breeds. A key gene, *IGF2*, is highly mitogenic, and it promotes the proliferation of various types of cells during the fetal period [31]. In fish, a report indicated the role of *IGF2* in the proliferation of spermatogonia and the inhibition of apoptosis [32]. In humans, treatment with an inhibitor of *IGF2* in spermatogonial stem cells decreased the proliferative activity of these germ cells [33]. Chicken *IGF2* is considered to be the most important candidate gene that can influence some growth traits, including growth, body measurement, and carcass [34]. However, there is little information available regarding the expression of *IGF2* and its relationship with germ cell proliferation and apoptosis during the reproductive cycle. The *E2F* transcription factor family has been proven to play an important role in cell cycle regulation. They regulate cell proliferation by regulating the transcription of target genes that are critical for the cell cycle [35]. Further studies on the *E2F* transcription factor family revealed a link between the function of the *E2F* transcription factor and the development of germ cells. It has been reported that the expression of *E2F* transcription factors is reduced in yak spermatogonia, thereby promoting the differentiation process of spermatogonia from mitotic to meiotic transition [36]. In mice, the findings indicate that *E2F6* possesses a broad ability to bind to and regulate the meiosis-specific gene population [37]. In the present study, we observed that *E2F4*, *E2F6*, and *IGF2*, the factors that promote cell proliferation, were significantly up-regulated in PGCs of You chicken and Hy-Line chicken at the E10 stage, which corresponded to the mitotic stage of rapid proliferation. *CDKN1B* and *BNIP3*, two key genes for apoptosis promotion [24], were significantly down-regulated in the PGCs of these two breeds of chicken at the E10 stage. It has been reported that *CDKN1B* and *BNIP3* can promote the apoptosis of chicken granulosa cells and trigger follicular atresia [38]. Upregulation of *CDKN1B* could inhibit cell cycle proteins, stopping the cell cycle at the G0–G1 transition, and thereby blocking cell division. Proapoptotic cytokines such as BNIP3, on the other hand, could induce apoptosis [39]. Therefore, we speculate that factors such as *E2F*, *IGF2*, *CDKN1B*, and *BNIP3* may be the potential signals leading to the rapid proliferation of PGCs in the two types of chickens.

The transition from mitosis to meiosis is a unique feature of germ cell development. Retinoic acid 8 (*STRA8*) is a pre-meiotic germ cell marker that is expressed by germ cells in response to retinoic acid (RA). In mice, it is reported that *STRA8* regulates the meiotic initiation of spermatogenesis and oogenesis [40]. The initiation of meiosis in the human ovary also requires the expression of *STRA8* [41]. In chickens, it has been reported that the expression of *STRA8* is crucial for the initiation of germ cell meiosis [25,42]. Another gene that is important for meiosis initiation, recurrent dehydrogenase 2 (*RALDH2*), has been extensively studied. In chickens, *RALDH2* is expressed in the left ovarian cortex at the time of *STRA8* up-regulation, before meiosis [25]. RA is the key factor controlling meiotic initiation in many animal species. In chicken embryos, *RALDH2* is the major enzyme responsible for RA synthesis, and sites of *RALDH2* gene expression correlate with chicken embryo RA production and release, as also observed in the mouse [43]. Furthermore, a *RALDH2* knockdown chicken model was produced to investigate the fundamental role of *RALDH2* in meiosis initiation. It was found that meiosis occurred abnormally in *RALDH2* knockdown ovaries [44]. Through transcriptome analysis, we found that the meiotic-promoting-related genes *STRA8* and *RALDH2* began to be significantly up-regulated in the PGCs of both breeds of chicken at the E14 stage. In addition, we speculate that methylation modification enzymes may be associated with the initiation of meiotic genes during germ cell differentiation. PGCs mainly undergo DNA demethylation during migration to meiosis [45]. Previous reports revealed that hydroxylation of 5-methylcytosines, mediated by *TET* proteins, participates in the active DNA demethylation of the zygotic paternal genome after fertilization. Further, both active and passive demethylation of the PGC genome is involved in expansion and migration [46,47,48]. The proteins of the *TET* family are not only involved in DNA demethylation during early embryonic development [46,49,50], but they are also critical for the initiation of meiosis in female germ cells [30]. Consistent with this, in our results, *TET2* was significantly up-regulated in PGCs of the two breeds of chicken at the E14 stage. However, further study on the mechanistic aspects of these genes during the germ cell development of birds is required.

In general, the broilers and layers exhibit the same transcriptome changes and share the majority of the differentially expressed genes during this transitional turning point in PGC development. However, the extent of up- or down-regulation of specific meiotic genes may show a subtle difference between the two breeds. We assume the differences in the magnitudes of the same directionally changing gene expression levels may lead to various degrees of meiosis implementation and execution, but they should not result in cell number variations. Therefore, the initiation of PGC meiosis is simultaneous between the two breeds, and future studies are necessary to fully unravel the molecular mechanisms regulating chicken PGC meiosis.

In conclusion, this study explored the dynamic and preserved transcriptome changes of PGCs from mitosis to meiosis in You chicken and Hy-Line chicken, understood the gene signatures that control the mitotic and meiotic activities of PGCs, elucidated the synchronous progression from mitosis to meiosis in PGCs between broilers and layers, and also provided theoretical guidance and an experimental platform for further research on the development of germ cells.

## Figures and Tables

**Figure 1 genes-14-00781-f001:**
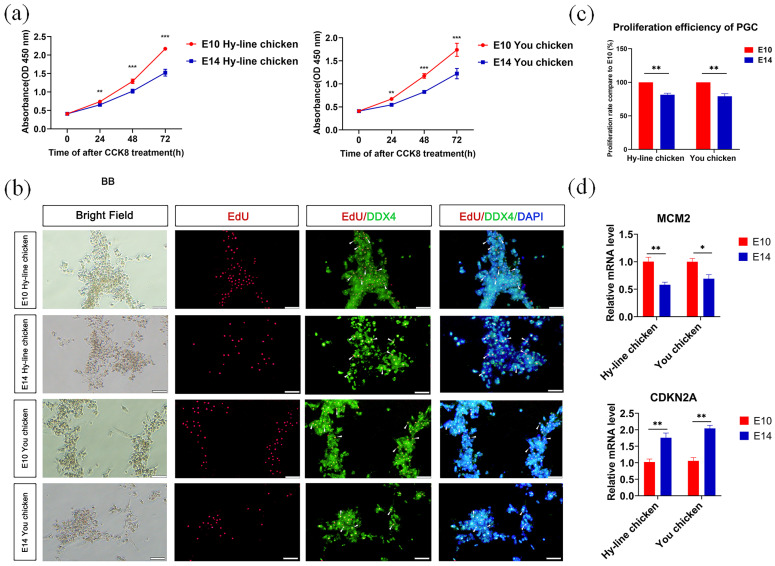
Comparison of the proliferation ability of E10 and E14 PGCs in both chicken breeds. (**a**) The CCK8 cell proliferation assay showed that the proliferation rate of PGCs gradually slowed down from E10 to E14 in both Hy-Line chicken (left) and You chicken (right). Data were presented as mean ± SEM, n = 5. *** *p* < 0.001. (**b**) Representative images of EdU staining and DDX4 immunostaining in cultured PGCs. The arrowheads indicate EdU-DDX4 double-positive cells. (**c**) Statistical analysis of the proliferation efficiency of PGCs. (**d**) The expression of proliferation-related genes was detected by RT-qPCR. RT-qPCR was performed with the 2^−ΔΔCt^ method for analysis. Data were presented as mean ± SEM, n = 5. * *p* < 0.05; ** *p* < 0.01; *** *p* < 0.001.

**Figure 2 genes-14-00781-f002:**
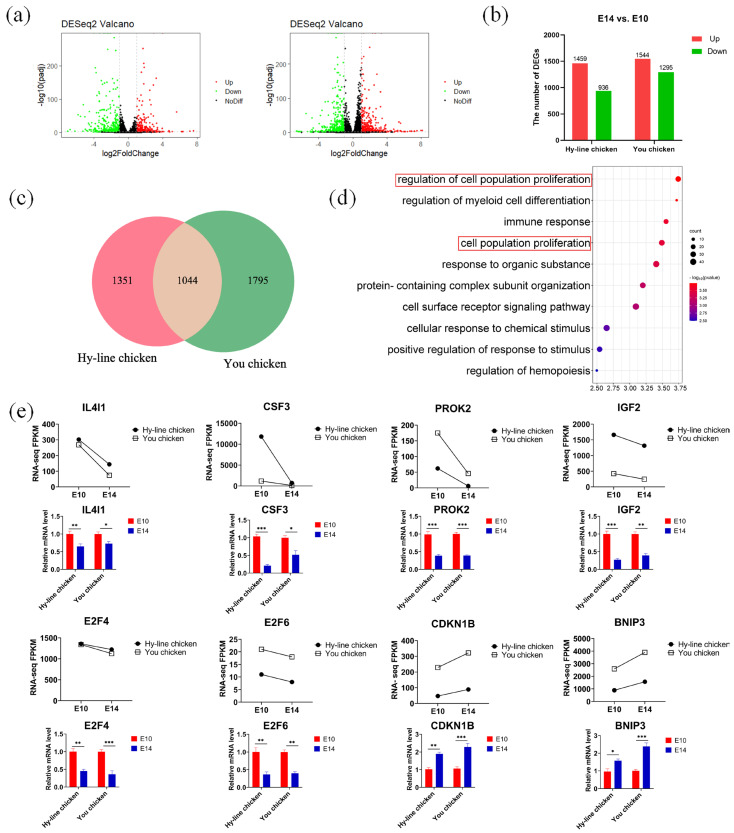
Differential co-expression genes involved in mitosis between Hy-Line chicken and You chicken. (**a**,**b**) Volcano map and histogram plots showing the differentially expressed genes between E10 PGCs and E14 PGCs, with the Hy-Line chicken group on the left and You chicken on the right. (**c**) The Venn diagram shows the differential genes co-expressed between the Hy-Line chicken and the You chicken groups. (**d**) Gene ontology terms enriched by 1044 DEGs. Colored shades represent *p*-values, and the size of the dot represents the number of genes. The redboxes indicate the enriched terms related to cell proliferation (**e**) Validation of key genes related to PGC proliferation by RT-qPCR. The line chart shows the change in FPKM values measured via RNA-Seq between E10 PGCs and E14 PGCs. The column chart shows the relative expression measured via RT-qPCR. Data were mean ± SEM, n = 5. * *p* < 0.05; ** *p* < 0.01; *** *p* < 0.001.

**Figure 3 genes-14-00781-f003:**
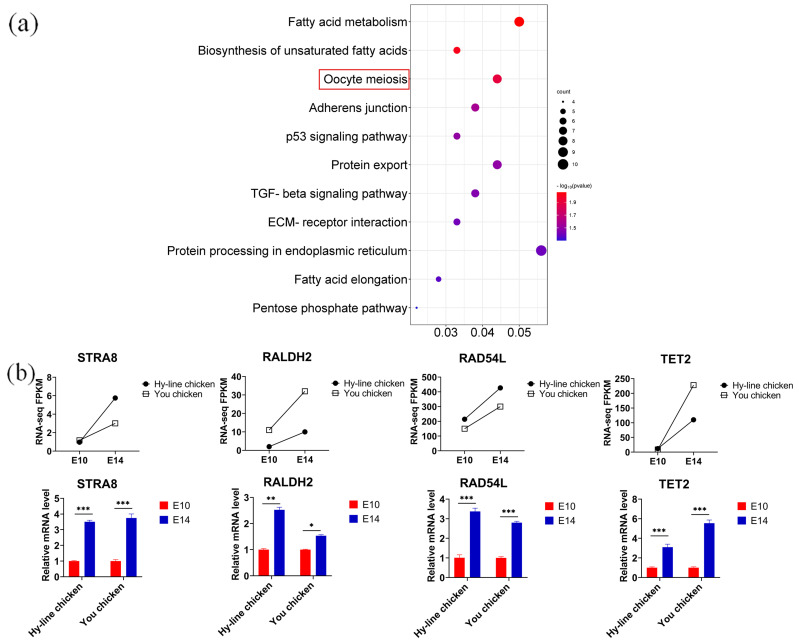
Differential co-expression genes involved in meiosis between Hy-Line chicken and You chicken. (**a**) KEGG pathway scatterplot enriched by co-upregulated DEGs in You chicken PGCs and Hy-Line chicken PGCs. The redbox indicates pathway related to meiosis. (**b**) RT-qPCR validation of selected genes related to meiosis initiation. The line chart shows the change in FPKM values measured via RNA-Seq between E10 PGCs and E14 PGCs. The column chart shows the relative expression measured via RT-qPCR. Data were presented as mean ± SEM, n = 5. * *p* < 0.05; ** *p* < 0.01; *** *p* < 0.001.

**Table 1 genes-14-00781-t001:** Genes regulating the process of cell proliferation.

Gene ID	log2 Fold Change (You Chicken)	log2 Fold Change (Hy-Line Chicken)	Gene Name
ENSGALG00010009474	4.637687	0.33777	*ACVR2A*
ENSGALG00010018099	0.640534	0.92462	*SMAD3*
ENSGALG00010015624	1.571854	0.709248	*ITGA1*
ENSGALG00010024658	−4.02396	−3.205	*CSF3*
ENSGALG00010023281	−0.39614	−0.84782	*B2M*
ENSGALG00010005131	−1.76463	−0.72073	*IL8L2*
ENSGALG00010024580	−0.33743	−0.40021	*IGF2*
ENSGALG00010028080	−0.84127	−0.85296	*MUSTN1*
ENSGALG00010005913	−1.23741	−0.51199	*NFKBIA*
ENSGALG00010020507	−0.7204	−2.24103	*ANKRD1*

**Table 2 genes-14-00781-t002:** Genes regulating “oocyte meiosis”.

Gene ID	log2 Fold Change (You Chicken)	log2 Fold Change (Hy-Line Chicken)	Gene Name
ENSGALG00010018388	0.331045	0.569627	*SMC3*
ENSGALG00010024756	1.115203	1.449926	*CDK2*
ENSGALG00000041923	0.730368	1.49584	*ESPL1*
ENSGALG00010017105	1.318177	3.994044	*PLK1*
ENSGALG00010022702	0.952322	3.161609	*CDC20*
ENSGALG00010017024	1.104456	3.314962	*BUB1*
ENSGALG00010018531	0.779517	0.531854	*ADCY6*

## Data Availability

The datasets generated and analyzed during the current study are available from the corresponding author upon reasonable request.

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
