# Peer review of "The Synchronized Progression from Mitosis to Meiosis in Female Primordial Germ Cells between Layers and Broilers"

_genes, 2023, doi:10.3390/genes14040781_

Round 1
Reviewer 1 Report
1. Please make the whole word and (abbreviation) in the first appearing word.
2. References No. 5, 12, 17, 18, 23, and 41. Please fill in all of the co-author names not uses et al.
Author Response
请参阅附件。

Reviewer 2 Report
In the manuscript titled "The synchronized progression from mitosis to meiosis in female primordial germ cells between layers and broilers", the authors use biocredit analysis to explore in depth the dynamic and conserved transcriptome changes of broiler and layer PGCs from mitosis to meiosis. It provides theoretical guidance and experimental platforms for further research on the development of germ cells, but some are not comprehensive and rigorous enough. Here are some of my suggestions:
1:Only the CCK-8 and EDU kits were used to detect the proliferation efficiency of the cells, qPCR or western blot experiments can be appropriately increased to detect proliferation-related gene expression.
2:Line 189, RNA-seq sample sequencing, why only two biological replicates were used at each developmental stage.
3:Line 260 "Genes such as STRA8, RALDH2, and RALD54L that play an important role in meiotic initiation", is there any literature to support it?
4:Line 266, Why focus on methylation modification genes, Is there more literature on the relationship between methylation modification genes and meiotic genes?
5:Some of the elements of the discussion section,it is recommended to put it in the introduction,for example, why choose Beijing You chicken and Hy-line chicken?
Author Response
请参阅附件。

Reviewer 3 Report
The introduction requires revision as it lacks clarity on the primary motivation and knowledge gap that prompted the authors to conduct the experiment. The inclusion of irrelevant information and sentences makes it difficult to grasp the main idea. As a research paper, it is important to be specific and direct. To improve the introduction, the authors should focus on providing a clear and concise statement of the research problem, the aim of the study, and the significance of their work. They should avoid unnecessary details and irrelevant information.
Development and optimization of chicken PGCs medium to culture and maintain PGCs was a critical step to allow progress in the study of PGCs biology. Please cite the original paper and not any random paper, i.e., reference 19.
Did the primers for the RT-PCR were designed or obtained from the available literature? Please add the needed information in the revised manuscript.
What did you use as a control to normalize or calibrate the mRNA expression of these genes on day 10 and day 14? For example, mRNA of day 8 old PGCs!!
Do you think two biological replicates for each development stage are statistically sound and reliable for RNA-Seq analysis?
Tables 1 and 2: please add them as supplementary data.
Figure 1: it is simply that you could successfully isolate and culture PGCs which is not the main objective of the study. Please add this figure as supplementary data.
Figure 2 c: please merge the proliferation rate of PGCs for both lines of chicken in the same figure.
Pearson's correlation coefficient (R) analysis between the replicates was performed in 2 replicates, right!!
In all cases figure 3 should also be included in the supplementary data and not in the main body of the manuscript.
Regarding Figure 4e, the presentation of relative mRNA expression in two different ways can be confusing for readers. To enhance clarity, the authors should eliminate the line graphs and keep only the bar figures. Additionally, it would be helpful to include one more figure that shows the comparison of mRNA expression levels of the eight genes between RT-PCR and RNA seq, providing evidence for the validity of RNA-Seq results. Similar recommendations should be applied to Figure 5.
While the discussion is concise and clear, the authors did not address the difference in gene expression between the two breeds adequately. It is important to provide a justification for why a particular gene shows a several-fold increase at day 14 in a specific breed compared to the other. Focusing on the functions of upregulated or downregulated genes alone is not sufficient, and the authors should elaborate on the implications of the differences in gene expression between the two breeds.
Author Response
请参阅附件。

Round 2
Reviewer 3 Report
The authors have addressed all the questions and incorporated the needed modifications in the revised manuscript.